# Lenvatinib Exacerbates the Decrease in Skeletal Muscle Mass in Patients with Hepatocellular Carcinoma, Whereas Atezolizumab Plus Bevacizumab Does Not

**DOI:** 10.3390/cancers16020442

**Published:** 2024-01-19

**Authors:** Kenji Imai, Koji Takai, Shinji Unome, Takao Miwa, Tatsunori Hanai, Atsushi Suetsugu, Masahito Shimizu

**Affiliations:** Department of Gastroenterology/Internal Medicine, Graduate School of Medicine, Gifu University, 1-1 Yanagido, Gifu 501-1194, Japan; takai.koji.t2@f.gifu-u.ac.jp (K.T.); unome.shinji.t7@f.gifu-u.ac.jp (S.U.); hanai.tatsunori.p8@f.gifu-u.ac.jp (T.H.); suetsugu.atsushi.e2@f.gifu-u.ac.jp (A.S.); shimizu.masahito.j1@f.gifu-u.ac.jp (M.S.)

**Keywords:** hepatocellular carcinoma, atezolizumab, bevacizumab, lenvatinib, prognostic factor, sarcopenia

## Abstract

**Simple Summary:**

Sarcopenia, the loss of skeletal muscle mass and strength, is associated with poor clinical outcomes in patients with hepatocellular carcinoma (HCC). This study indicated that lenvatinib treatment significantly exacerbated the decrease in skeletal muscle mass, whereas atezolizumab plus bevacizumab (AB) treatment did not. Furthermore, skeletal muscle mass was found to be an independent predictor of survival, together with alpha-fetoprotein and albumin-bilirubin scores, using time-varying covariates in the Cox proportional hazards model. These findings suggest that maintaining a relatively high skeletal muscle mass during AB treatment is advantageous for improving the prognosis of HCC chemotherapy.

**Abstract:**

This study aimed to evaluate chronological changes in skeletal muscle index (SMI), subcutaneous and visceral adipose tissue indices (SATI and VATI), AFP, PIVKA-II, and ALBI scores during atezolizumab plus bevacizumab (AB) or lenvatinib (LEN) treatment for hepatocellular carcinoma (HCC) and the effect of these changes on survival. A total of 94 patients with HCC (37 were on AB and 57 on LEN) were enrolled. SMI, SATI, VATI, AFP, PIVKA-II, and ALBI scores were analyzed at the time of the treatment introduction (Intro), 3 months after the introduction (3M), at drug discontinuation (End), and the last observational time (Last). The differences between chronological changes were analyzed using the Wilcoxon paired test. The independent predictors for survival and the changes in SMI during AB or LEN (c-SMI%) were analyzed using the Cox proportional hazards model treating all these factors as time-varying covariates and the analysis of covariance, respectively. SMI in the AB group was maintained over time (42.9–44.0–40.6–44.2 cm^2^/m^2^), whereas that in the LEN group significantly decreased during the Intro–3M (*p* < 0.05) and 3M–End (*p* < 0.05) period (46.5–45.1–42.8–42.1 cm^2^/m^2^). SMI (*p* < 0.001) was an independent predictor for survival together with AFP (*p* = 0.004) and ALBI score (*p* < 0.001). Drug choice (AB or LEN; *p* = 0.038) and PIVKA-II (*p* < 0.001) were extracted as independent predictors for c-SMI%. AB treatment was significantly superior to LEN in terms of maintaining skeletal muscle, which is an independent predictor for survival.

## 1. Introduction

Hepatocellular carcinoma (HCC) is one of the most common malignancies worldwide with a poor prognosis [1]. HCC is typically diagnosed after having progressed to an unresectable state [2], and despite curative treatment for the primary HCC, recurrence is common [3,4,5]. Therefore, most HCCs are unresectable during treatment and require subsequent therapy. According to the clinical guidelines for HCC in the US, Europe, and Japan [6,7,8], systemic chemotherapy is generally recommended for patients with preserved liver function and multinodular HCC accompanied by macrovascular invasion and/or metastatic disease. Moreover, systemic chemotherapy is increasingly introduced for single or multinodular HCC without macrovascular invasion or metastatic disease, which were previously treated using transcatheter arterial chemoembolization, in expectation of a high antitumor effect and preservation of liver function during chemotherapy [9,10,11].

Sorafenib is the first oral active multikinase inhibitor that prolongs survival by 3 months in patients with unresectable HCC compared to placebo [9]. After that, the REFLECT trial demonstrated that lenvatinib (LEN), also an oral multikinase inhibitor, targeting vascular endothelial growth factor (VEGF) receptors 1–3, fibroblast growth factor (FGF) receptors 1–4, and platelet-derived growth factor (PDGF) receptor α, was non-inferior to sorafenib in overall survival (OS) in untreated advanced HCC [10]. The IMbrave150 trial demonstrated that the combination of atezolizumab, a programmed death-ligand 1 (PD-L1)-targeted antibody, and bevacizumab, a monoclonal antibody that targets VEGF, resulted in better OS and progression-free survival (PFS) than sorafenib [11]. According to the latest guidelines for HCC [12], atezolizumab plus bevacizumab (AB) is currently the first-line chemotherapy for HCC, whereas LEN is the second line or the first line in cases where immunotherapy is contraindicated, such as in those with underlying autoimmune diseases.

Both AB and LEN are widely used as chemotherapeutic options for unresectable HCC; however, there is no direct head-to-head comparison between these treatments and it is debatable as to which treatment would be better in terms of the antitumor effect, survival benefit, and severity of adverse events (AEs). On comparing the IMbrave150 and REFLECT trials [10,11], the response rate and PFS for AB and LEN are similar (27.3% vs. 24.1% and 6.8 vs. 7.4 months, respectively), whereas AB seems a little superior to LEN in terms of OS (19.2 vs. 13.6 months). There were few differences in the frequencies of AEs at grade ≥3 between AB and LEN (56.5% vs. 57%); however, general fatigue, decreased appetite, and decreased weight at grade ≥3 were more frequent in LEN (2.4% vs. 4%, 1.2% vs. 5%, and 0% vs. 8%, respectively). Therefore, AE-related changes in body composition may have occurred, especially in the LEN-treated patients.

Sarcopenia is a progressive and generalized skeletal muscle disorder characterized by loss of skeletal muscle mass and strength [13]. Unfavorable changes in body composition such as sarcopenia or adipopenia can lead to poor survival in patients with HCC. Skeletal muscle depletion can predict poor prognosis in patients with HCC [14,15]. Rapid depletion of subcutaneous adipose tissue and skeletal muscle is associated with poor survival in patients with HCC treated with sorafenib [16]. Furthermore, skeletal muscle mass decreases significantly during LEN or sorafenib treatment for unresectable HCC and is an independent prognostic factor as indicated by using survival analysis with time-varying covariates [17]. These findings suggest that the advantages of AB that maintain the activity and body weight of the patients could prevent sarcopenia and/or adipopenia and eventually contribute to survival benefits.

In this study, we measured the chronological changes in skeletal muscle and subcutaneous and visceral adipose tissue, together with the tumor markers for HCC and liver functional reserve, from the time of introduction of AB or LEN to the last observational period and clarified whether these changes would affect survival in patients with advanced HCC by conducting a survival analysis with time-varying covariates.

## 2. Materials and Methods

### 2.1. Enrolled Patients and Treatment Strategy

A total of 94 patients with HCC undergoing treatment with AB (*n* = 37) and LEN (*n* = 57) between April 2018 and December 2022 at the Gifu University Hospital were enrolled in this study. We selected AB and LEN treatments according to the latest guidelines for HCC [12]. Patients assigned to the AB group received 1200 mg of atezolizumab plus 15 mg per kilogram of body weight of bevacizumab intravenously every 3 weeks [11], whereas LEN was administered at a dose of 12 mg/day (for body weight ≥60 kg) or 8 mg/day (for body weight <60 kg) [10]. Dose modifications were not permitted in the AB group; however, they were allowed in the LEN group. In our hospital, dynamic computed tomography (CT) was performed every 3 or 4 months during chemotherapy to evaluate the treatment response, which was measured according to the modified RECIST [18]. AEs were assessed according to the Common Terminology Criteria for Adverse Events, version 5.0. OS and PFS were defined as the time from the date of the first AB or LEN treatment to the date of death and progressive disease (PD), respectively. Unacceptable AEs or the deterioration of liver functional reserve and PD were observed, and a change in the treatment strategies for HCC was considered; however, AB or LEN could be continued beyond PD as long as there was a clinical benefit.

Since this is a retrospective study, the participants in this study did not provide written informed consent. Instead, we provided all participants with an opportunity to opt out by disclosing the details of the study. The study design, including this consent procedure, was approved by the ethics committee of the Gifu University School of Medicine on 2 June 2021 (ethical protocol code: 29–26).

### 2.2. Image Analysis of Body Composition

We determined the skeletal muscle index (SMI, cm^2^/m^2^) by normalizing the cross-sectional areas of the skeletal muscle (cm^2^) at the third lumbar vertebra on the CT image by the square of the patient’s height (m^2^) using the SYNAPSE VINCENT software (version 6.7, Fujifilm Medical, Tokyo, Japan) [19]. In the same manner, the subcutaneous adipose tissue index (SATI, cm^2^/m^2^) and the visceral adipose tissue index (VATI, cm^2^/m^2^) were determined from the CT cross-sectional areas of the subcutaneous and visceral adipose tissues at the level of the umbilicus, respectively. SMI, SATI, and VATI, together with alpha-fetoprotein (AFP) and proteins induced by vitamin K absence or antagonist-II (PIVKA-II), tumor markers for HCC, and the ALBI score, an indicator for liver functional reserve, were collected a total of four times during this study: at the time of AB or LEN introduction (Intro), three months after the introduction (3M), at the drug discontinuation (End), and at the last observational time (Last).

### 2.3. Statistical Analysis

Baseline characteristics were compared using Student’s *t*-test for continuous variables or Fisher’s exact test for categorical variables. The Pearson product–moment correlation coefficient was used for measuring the linear correlation between two continuous variables. OS and PFS were estimated using the Kaplan–Meier method and the differences between the curves were evaluated using a log-rank test. Differences between chronological changes in the SMI, SATI, VATI, AFP, PIVKA-II, and ALBI scores were analyzed using the Wilcoxon paired test. Additionally, we evaluated the changes in SMI, SATI, and VATI during the initial treatment and the introduction of AB or LEN to estimate the effect of pretreatment on body composition. We designated all parameters as time-varying covariates [20] and selected the independent predictors for OS among them using the Cox proportional hazards model. Furthermore, c-SMI%, the percentage of the changes in SMI during AB or LEN treatment, was defined as follows: c-SMI% = SMI at the time of the discontinuation ×100/SMI at the time of the introduction. The analysis of covariance (ANCOVA) was used to determine which factors including age, sex, liver functional reserve, tumor-related factors, and the severities of AEs would affect c-SMI%. *p* < 0.05 was considered statistically significant. All statistical analyses were performed using R software ver. 4.2.2 (R Foundation for Statistical Computing, Vienna, Austria; http://www.R-project.org/ (accessed on 30 November 2023)).

## 3. Results

### 3.1. Baseline Demographics and Clinical Characteristics of the Enrolled Patients

The baseline demographics and clinical characteristics of the enrolled patients (37 on AB and 57 on LEN) immediately before the time of AB or LEN introduction are presented in Table 1. There were no significant differences except for the Barcelona Clinic Liver Cancer stage (*p* = 0.006) between the two groups. Esophagogastroduodenoscopy was conducted annually in patients with esophageal varices and endoscopic variceal ligation was performed for esophageal varices with more than F2 and/or a red color sign. Only one patient (LEN group) experienced bleeding from the varices. The information associated with pre-, combination-, and post-treatment of the enrolled patients is detailed in Appendix A. A total of thirty-three (89.2%) and fifty-one (89.5%) patients treated with AB and LEN underwent pre-treatments, four (10.8%) and fourteen (24.6%) patients received combination treatments during AB or LEN, and fifteen (40.5%) and twenty-eight (49.1%) patients received post-treatments, respectively. 

### 3.2. Treatment Responses and Adverse Events in the Enrolled Patients

The OS rates at 1 and 2 years and the median OS of the patients treated with AB and LEN were 79.0%, 40.5%, and 20.8 months, and 51.6%, 32.3%, and 12.1 months, respectively (Figure 1a). PFS rates at 1 and 2 years and the median PFS were 27.4%, 17.6%, and 7.6 months, and 39.0%, 17.5%, and 7.8 months, respectively (Figure 1b). The treatment responses of complete response (CR), partial response (PR), stable disease (SD), and PD in patients treated with AB and LEN were observed in two, eleven, fifteen, and nine cases and five, eleven, seventeen, and twenty cases, respectively (Table 1). The objective response and disease control rates of the AB and LEN groups were 35.1% and 75.7%, and 30.2% and 62.2%, respectively. There were no significant differences between the two groups in terms of OS, PFS, objective response, and disease control rate.

Table 2 lists the AEs recorded in response to AB or LEN. We found that 36 (97.3%) and 55 (96.5%) patients experienced some form of AEs at any grade, and 15 (40.5%) and 19 (33.3%) at grade ≥ 3 in the AB and LEN groups, respectively. Decreased appetite (52.6% vs. 13.5%) and general fatigue (52.7% vs. 8.1%) at grade ≥ 2 were more often observed in the LEN group than in the AB group. No patients in the AB group experienced hand–foot syndrome; however, six experienced immune-related AEs including interstitial pneumonia, rush, neuropathy, hypopituitarism, and hepatitis. Grade 5 events did not occur in either group.

### 3.3. Chronological Changes in SMI, SATI, VATI, AFP, PIVKA-II, and ALBI Scores

The chronological changes in SMI at Intro, 3M, End, and Last were 42.9–44.0–40.6–44.2 cm^2^/m^2^ for the AB group and 46.5–45.1–42.8–42.1 cm^2^/m^2^ for the LEN group (Figure 2a). SMI in the AB group was maintained during this study, whereas that in the LEN group significantly decreased during the Intro–3M and 3M–End periods (*p* < 0.05, respectively). As for SATI (Figure 2b) and VATI (Figure 2c), VATI in the LEN group during the 3M–End period significantly decreased (*p* < 0.05). Changes in SMI, SATI, and VATI during the initial treatment and the introduction of AB were 43.2–42.9 cm^2^/m^2^ (*p* = 0.777), 50.0–42.8 cm^2^/m^2^ (*p* = 0.119), and 44.7–54.7 cm^2^/m^2^ (*p* = 0.002), respectively, whereas those of LEN were 45.0–46.5 cm^2^/m^2^ (*p* = 0.931), 47.1–52.1 cm^2^/m^2^ (*p* = 0.164), and 47.0–48.8 cm^2^/m^2^ (*p* = 0.891), as shown in Appendix A.

Chronological changes in AFP, PIVKA-II, and ALBI scores are shown in Figure 3. AFP level significantly increased during the 3M–End period for AB (*p* < 0.05) and the End–Last period for LEN (*p* < 0.05, Figure 3a). PIVKA-II level significantly increased during the Intro–3M, 3M–End, and End–Last period for LEN (*p* < 0.05, respectively; Figure 3b). The ALBI score significantly deteriorated during the Intro–3M period for AB (*p* < 0.05) and during the Intro–3M and 3M–End periods for LEN (*p* < 0.05, respectively; Figure 3c).

### 3.4. Analysis of the Independent Predictors for Survival and c-SMI%

Table 3 shows the results of survival analysis, which permits the selection of independent predictors for OS using the Cox proportional hazards model, with all variables designated as time-varying covariates. This demonstrated that AFP (hazard ratio (HR): 1.014, 95% confidence interval (CI): 1.004–1.024, *p* = 0.004), ALBI score (HR: 3.213, 95% CI: 1.997–5.168, *p* < 0.001), and SMI (HR: 0.942, 95% CI: 0.913–0.972, *p* < 0.001) were independent predictors for survival.

The average c-SMI% of all the patients was 93.4% and those divided by treatment differences and the presence of decreased appetite and general fatigue at grade <2 and grade ≥2 were 97.9% vs. 91.0% (*p* = 0.030, Appendix A), 96.3% vs. 89.0% (*p* = 0.015, Appendix A), and 95.6% vs. 89.6% (*p* = 0.054, Appendix A), respectively. PIVKA-II at the introduction was also negatively correlated with c-SMI% (correlation coefficient = −0.414, *p* < 0.001, Appendix A). Among them, the results of the ANCOVA revealed that drug (*p* = 0.038) and PIVKA-II (*p* < 0.001) were chosen as the independent predictors for c-SMI% (Table 4). 

## 4. Discussion

This study demonstrated that patients with HCC undergoing treatment with LEN continued to significantly lose their skeletal muscle mass, whereas those treated with AB did not. Decreased appetite and general fatigue at grade ≥2 were more often observed in the LEN group than that in the AB group and the patients who experienced these AEs tended to lose their skeletal muscle mass. In contrast, the decrease in skeletal mass was significantly higher in patients treated with LEN than in those treated with AB even after the effect of the severity of decreased appetite was adjusted. Furthermore, pretreatment with hepatectomy, radiofrequency ablation, or transcatheter arterial chemoembolization did not reduce the skeletal muscle mass in either group. These results imply that LEN might also directly decrease the skeletal muscle mass regardless of the characteristics of the patients, tumor factors, and the severities of AEs.

LEN exerts a high antitumor effect by inhibiting multiple tyrosine kinase receptors such as VEGF, FGF, and PDGF receptors involved in carcinogenesis [10]. Additionally, these receptors have been reportedly involved in muscle protein synthesis [21,22,23]. All these tyrosine kinase receptors are associated with the activation of the PI3K/AKT/mTOR pathway, a key regulator of muscle protein synthesis. Therefore, tyrosine kinase inhibitors such as LEN and sorafenib could exacerbate decreased skeletal muscle by inhibiting this signaling pathway [24]. Bevacizumab is also a monoclonal antibody that targets VEGF [11]; skeletal muscle mass was relatively preserved in the AB group in this study. These findings suggest that inhibiting multiple tyrosine kinase receptors may exert a stronger effect on skeletal muscle mass reduction than targeting a single receptor; however, elucidating these details requires further investigation.

Patients with liver disease including those with HCC are more likely to be complicated by sarcopenia, which is associated with poor clinical outcomes [13]. The Cox proportional hazards model using time-varying covariates conducted in this study demonstrated that the volume of skeletal muscle in patients with HCC undergoing treatment with AB or LEN was one of the independent predictors for survival together with AFP and ALBI score. These results align with previous studies [14,15,17]. Therefore, in addition to reducing tumor burden and maintaining liver function reserve to the maximum extent possible, skeletal muscle mass, which tends to decrease during chemotherapy, especially tyrosine kinase inhibitor (TKI) treatment, must be maintained to improve the survival of patients with unresectable HCC. Several methods are used in clinical practice to measure body composition, including skinfold thickness assessment, bioelectrical impedance analysis, dual-energy X-ray absorptiometry, CT (used in this study), and quantitative magnetic resonance imaging [25]. Among them, we believe that CT is the most suitable assessment technique for patients with HCC because it is already commonly used to evaluate the treatment response for HCC. Interestingly, with respect to tumor markers, AFP was one of the independent predictors for survival, whereas PIVKA-II was one for the changes in SMI during AB or LEN. Thus, both AFP and PIVKA-II should be measured during AB or LEN treatments to evaluate the antitumor effect and predict the changes in SMI. It is essential to reduce AFP and PIVKA-II and maintain SMI and ALBI scores to the maximum to improve survival in patients with HCC that are undergoing treatment with AB or LEN.

To maintain the volume of skeletal muscle in patients with HCC undergoing AB or LEN treatment, it is essential to practice adequate nutritional and exercise therapies. If decreased appetite comes from AEs of chemotherapy, consideration should be given to reducing the dose of TKIs or changing the treatment strategy for HCC. In this study, decreased appetite at grade ≥2 was more often observed in the LEN group than in the AB group. Therefore, when the AE of decreased appetite in patients who are treated with LEN and have not yet undergone AB is intolerable, it is quite reasonable to switch from LEN to AB. Branched-chain amino acid (BCAA) supplementation is a promising nutritional strategy to maintain skeletal muscle and improve survival in patients with liver diseases [26,27]. Among patients with HCC undergoing sorafenib treatment, those who received BCAAs had longer survival and higher serum levels of albumin than the control group [28]. Exercise therapy, which is also strongly recommended for patients with cancer [29], can increase skeletal muscle mass and improve survival in patients with HCC [30]. Furthermore, BCAA supplementation and exercise therapy can synergistically prevent skeletal muscle atrophy in patients with HCC [31]. Additionally, nutrition and exercise therapies improve pathological conditions such as obesity and diabetes mellitus, which can promote carcinogenesis in the liver and worsen survival in patients with HCC [32,33]. Therefore, both malnutrition and overnutrition must be addressed at all times, and efforts must be made to maintain adequate body composition through appropriate nutritional and exercise therapies.

This study had several limitations. First, this was a retrospective single-center study with a comparatively small sample size. Second, 10.8% of the AB group and 24.6% of the LEN group underwent other combination treatments. Thus, we could not estimate the isolated effects of AB or LEN on body composition, liver functional reserve, and therapeutic response. Furthermore, most patients underwent pre-, combination, or post-treatment for HCC, which must have affected OS. To clarify the pure effects of AB or LEN on body composition, liver functional reserve, AEs, and OS, a prospective study involving a larger number of patients enrolled from several centers should be performed.

## 5. Conclusions

Patients with HCC who had undergone LEN treatment continued to significantly lose their skeletal muscle mass, whereas those who had undergone AB treatment did not. Skeletal muscle mass was found to be one of the independent predictors for survival together with AFP and ALBI score, using time-varying covariates in the Cox proportional hazards model. Furthermore, the choice of the drugs (AB or LEN) and PIVKA-II were chosen as independent predictors for the changes in SMI during the AB or LEN treatment. The ability to maintain relatively high skeletal muscle mass during AB treatment is advantageous for improving the prognosis of HCC chemotherapy.

## Figures and Tables

**Figure 1 cancers-16-00442-f001:**
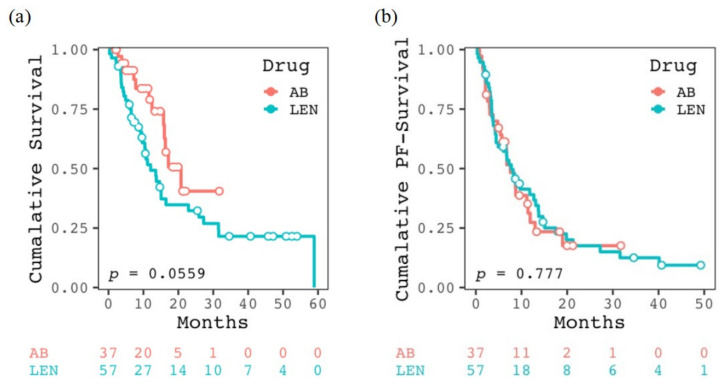
Kaplan–Meier curves for (**a**) cumulative survival and (**b**) cumulative progression-free (PF) survival in the two groups treated with atezolizumab plus bevacizumab (AB) and lenvatinib (LEN).

**Figure 2 cancers-16-00442-f002:**
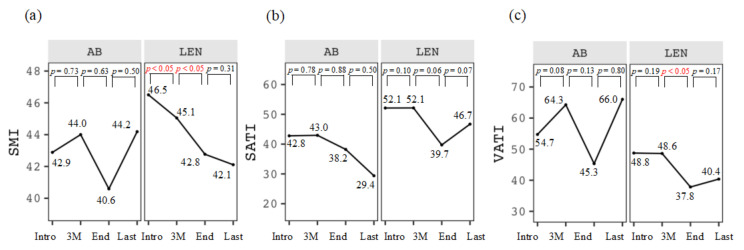
Chronological changes in (**a**) skeletal muscle index (SMI, cm^2^/m^2^), (**b**) subcutaneous adipose tissue index (SATI, cm^2^/m^2^), and (**c**) visceral adipose tissue index (VATI, cm^2^/m^2^) at the introduction (Intro), 3 months after the introduction (3M), at the discontinuation (End), and the last observational time (Last) in the two groups treated with atezolizumab plus bevacizumab (AB) and lenvatinib (LEN).

**Figure 3 cancers-16-00442-f003:**
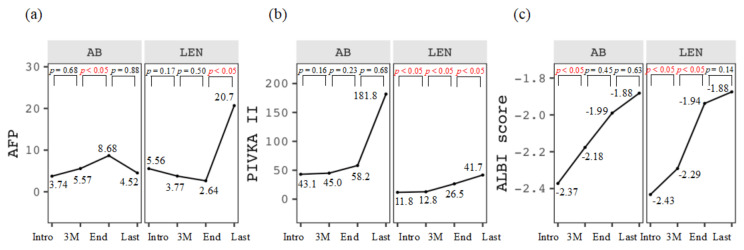
Chronological changes in (**a**) AFP (×10^3^ ng/mL), (**b**) PIVKA-II (×10^3^ mAU/mL), and (**c**) ALBI score at the introduction (Intro), three months after the introduction (3M), at the discontinuation (end), and the last observational time (Last) in the two groups treated with atezolizumab plus bevacizumab (AB) and lenvatinib (LEN).

**Table 1 cancers-16-00442-t001:** Baseline demographic and clinical characteristics of the enrolled patients.

Valuables	All Patients (*n* = 94)	AB Group (*n* = 37)	LEN Group (*n* = 57)	*p*-Value
Age (years)	75 (68.5–81)	76 (72–82)	75 (68–80)	0.272
Sex (male/female)	73/21	28/9	45/12	0.801
Etiology (HBV/HCV/others)	16/22/56	5/11/21	11/11/35	0.487
BCLC stage (A/B1/B2/C)	7/9/21/57	6/4/11/16	1/5/10/41	0.006
Child–Pugh score (5/6/7/8/9)	53/32/7/1/1	21/16/0/0/0	32/16/7/1/1	0.069
ALBI score	−2.460 (−2.80–2.06)	−2.420 (−2.62–2.13)	−2.52 (−2.86–2.05)	0.566
Esophageal varices (no/F1/F2/F3 post EVL)	52/14/1/2/15	23/5/0/0/7	29/9/1/2/8	0.792
SMI (cm^2^/m^2^)	43.6 (38.7–50.5)	41.7 (36.4–47.7)	44.5 (40.6–50.8)	0.123
SATI (cm^2^/m^2^)	40.4 (29.2–67.0)	38.1 (30.4–53.8)	44.7 (27.9–67.6)	0.194
VATI (cm^2^/m^2^)	50.5 (32.6–66.3)	56.7 (36.1–67.2)	46.6 (29.0–64.2)	0.325
ALB (g/dL)	3.7 (3.4–4.1)	3.7 (3.5–3.9)	3.9 (3.4–4.3)	0.362
T-Bil (mg/dL)	0.8 (0.6–1.1)	0.7 (0.6–1.0)	0.8 (0.7–1.1)	0.236
AFP (×10^3^ ng/mL)	0.169 (0.009–2.470)	0.098 (0.008–1.618)	0.225 (0.010–2.770)	0.061
PIVKA-II (×10^3^ mAU/mL)	0.758 (0.119–5.935)	0.640 (0.092–5.518)	0.917 (0.220–6.069)	0.265
Best response (CR/PR/SD/PD)	7/22/32/29	2/11/15/9	5/11/17/20	0.443

Continuous covariates are presented as median (interquartile range). HBV, hepatitis B virus; HCV, hepatitis C virus; BCLC stage, Barcelona Clinic Liver Cancer stage; EVL, endoscopic variceal ligation; SMI, skeletal muscle index; SATI, subcutaneous adipose tissue index; VATI, visceral adipose tissue index; ALB, albumin; T-Bil, total bilirubin; AFP, alpha-fetoprotein; PIVKA-II, protein induced by vitamin K absence or antagonist-II; CR, complete response; PR, partial response; SD, stable disease; PD, progressive disease.

**Table 2 cancers-16-00442-t002:** Adverse events during chemotherapy.

	AB Group (*n* = 37)				LEN Group (*n* = 57)			
	Any Grade	Grade 1	Grade 2	Grade ≥ 3	Any Grade	Grade 1	Grade 2	Grade ≥ 3
Any symptoms	36 (97.3%)			15 (40.5%)	55 (96.5%)			19 (33.3%)
Decreased appetite	12 (32.4%)	7 (18.9%)	4 (10.8%)	1 (2.7%)	33 (57.9%)	3 (5.3%)	28 (49.1%)	2 (3.5%)
General fatigue	15 (40.5%)	12 (32.4%)	3 (8.1%)	0	32 (56.1%)	2 (3.5%)	27 (47.4%)	3 (5.3%)
Hypertension	15 (40.5%)	2 (5.4%)	9 (24.3%)	4 (10.8%)	23 (40.4%)	2 (3.5%)	14 (24.6%)	7 (12.3%)
Proteinuria	19 (51.4%)	8 (21.6%)	5 (13.5%)	6 (16.2%)	18 (31.6%)	4 (7.0%)	9 (15.8%)	5 (8.8%)
Hand–foot syndrome	0	0	0	0	17 (29.8%)	7 (12.3%)	10 (17.5%)	0
Hypothyroidism	13 (35.1%)	9 (24.3%)	4 (10.8%)	0	17 (29.8%)	0	17 (29.8%)	0
Diarrhea	8 (21.6%)	7 (18.9%)	0	1 (2.7%)	13 (22.8%)	11 (19.3%)	1 (1.8%)	1 (1.8%)
Liver dysfunction	17 (45.9%)	16 (43.2%)	0	1 (2.7%)	7 (12.3%)	1 (1.8%)	5 (8.8%)	1 (1.8%)
Hemorrhage	3 (8.1%)	3 (8.1%)	0	0	3 (5.3%)	0	3 (5.3%)	0

**Table 3 cancers-16-00442-t003:** Univariate and multivariate analyses of predictors for survival using the Cox proportional hazards model.

Variable	Univariate Analysis		Multivariate Analysis	
	HR (95%CI)	*p*-Value	HR (95%CI)	*p*-Value
AFP (×10^3^ ng/mL)	1.018 (1.009–1.026)	<0.001	1.014 (1.004–1.024)	0.004
PIVKA-II (×10^3^ mAU/mL)	1.001 (0.999–1.003)	0.240		
BCLC stage A vs. B1	0.342 (0.021–5.520)	0.450		
vs. B2	2.271 (0.292–17.65)	0.433		
vs. C	3.082 (0.420–22.61)	0.268		
vs. D	4.581 (0.285–73.68)	0.283		
ALBI score	3.158 (2.123–4.697)	<0.001	3.213 (1.997–5.168)	<0.001
SMI (cm^2^/m^2^)	0.954 (0.924–0.985)	0.004	0.942 (0.913–0.972)	<0.001
SATI (cm^2^/m^2^)	0.997 (0.987–1.007)	0.569		
VATI (cm^2^/m^2^)	0.995 (0.983–1.006)	0.357		

All the variables except BCLC stage were dealt with as time-varying covariates. HR, hazard ratio; CI, confidence interval; AFP, alpha-fetoprotein; PIVKA-II, protein induced by vitamin K absence or antagonist-II; ALBI, albumin-bilirubin; BCLC, Barcelona Clinic Liver Cancer; SMI, skeletal muscle index; SATI, subcutaneous adipose tissue index; VATI, visceral adipose tissue index.

**Table 4 cancers-16-00442-t004:** ANCOVA results for factors affecting the changes in SMI during AB or LEN treatment.

Variable	Univariate Analysis		Multivariate Analysis	
	PRC	*p*-Value	PRC	*p*-Value
Age	0.025	0.874		
SEX (male versus female)	−0.261	0.944		
Drug (LEN versus AB)	−6.841	0.030	−6.290	0.038
ALBI score	−0.397	0.901		
SATI (cm^2^/m^2^)	0.068	0.138		
VATI (cm^2^/m^2^)	0.017	0.782		
AFP (×10^3^ ng/mL)	−0.207	0.104		
PIVKA-II (×10^3^ mAU/mL)	−0.034	<0.001	−0.038	<0.001
Decreased appetite (≥G2 versus <G2)	−7.369	0.015	−5.418	0.061
General fatigue (≥G2 versus <G2)	−5.986	0.054		
Hypertension (≥G2 versus <G2)	0.925	0.763		
Proteinuria (≥G2 versus <G2)	0.227	0.949		
Hypothyroidism (≥G2 versus <G2)	−2.364	0.504		
Diarrhea (≥G2 versus <G2)	10.636	0.241		
Best response (SD/PD versus CR/PR)	−2.765	0.379		

ANCOVA, analysis of covariance; SMI, skeletal muscle index; AB, atezolizumab plus bevacizumab; LEN. lenvatinib; PRC, partial regression coefficient; ALBI, albumin-bilirubin; SATI, subcutaneous adipose tissue index; VATI, visceral adipose tissue index; AFP, alpha-fetoprotein; PIVKA-II, protein induced by vitamin K absence or antagonist-II; CR, complete response; PR, partial response; SD, stable disease; PD, progressive disease.

## Data Availability

The data presented in this study are available upon request from the corresponding author.

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
