# Peer review of "Lenvatinib Exacerbates the Decrease in Skeletal Muscle Mass in Patients with Hepatocellular Carcinoma, Whereas Atezolizumab Plus Bevacizumab Does Not"

_cancers, 2024, doi:10.3390/cancers16020442_

Round 1
Reviewer 1 Report
Comments and Suggestions for Authors
Please, review all material and methods and put other analyses as exercises and previous treatments to prevent loss weight in the 2 groups described. This point can be discussed too.
Comments on the Quality of English LanguagePlease, review the language in all text. There are minor problems.
The principal suggestion is to modify the title. The title is longer than necessary and it is not objective.
Author Response
Response to Reviewer #1
We are pleased that in the overall comments this reviewer found our study is of interest. We also thank this reviewer’s constructive comments which were most helpful to improve our manuscript. We accordingly revised the manuscript as follows.
- Please, review all material and methods and put other analyses as exercises and previous treatments to prevent loss weight in the 2 groups described. This point can be discussed too.
I am very sorry to inform you that we do not have chronological data of exercises and body weight over time because our study is retrospective. Instead, we evaluated the changes in SMI, SATI, and VATI during the initial treatment and the introduction of AB or LEN to estimate the effect of pre-treatment on body composition. The result demonstrated that changes in SMI, SATI, VATI during the initial treatment and the introduction of AB were 43.2–42.9 cm2/m2 (p = 0.777), 50.0–42.8 cm2/m2 (p = 0.119), and 44.7–54.7 cm2/m2 (p = 0.002), whereas those of LEN were 45.0–46.5 cm2/m2 (p = 0.931), 47.1–52.1 cm2/m2 (p = 0.164), and 47.0–48.8 cm2/m2 (p = 0.891) as shown in Table S2. This fact revealed that pre-treatment such as hepatectomy, radiofrequency ablation and transcatheter arterial chemoembolization did not reduce skeletal muscle mass in both groups, which also strengthen our suggestion that TKIs including lenvatinib tend to decrease skeletal muscle mass compared with other treatments for HCC. We added these revisions in the Materials and Methods section (Page 3, lines 39-41), Results section (Page 5, lines 30 to Page 6 lines 2), Discussion section (Page 8, lines 1-3) with new Table S2. These revisions contribute significantly to the paper's quality. We appreciate your valuable suggestion.
- Please, review the language in all text. There are minor problems.
We have already asked English editing by a native English speaker. We also submit the English proofread certificate for this reviewer.
- The principal suggestion is to modify the title. The title is longer than necessary and it is not objective.
According to this suggestion, we changed the title of this study as follow: Lenvatinib continues to decrease skeletal muscle mass in patients with hepatocellular carcinoma, whereas atezolizumab plus bevacizumab does not. We appreciate your valuable suggestion again.

Reviewer 2 Report
Comments and Suggestions for Authors
In this retrospective study Smai et al showed that in patients with hepatocellular carcinoma (HCC) treatment with Lenvatinib (LEN) induced loss of muscular mass (SMI), while atezolizumab + bevacizumab (AB) did not. Furthermore, they confirmed that SMI was a survival predictor in combination with ALBI index. Main comments:
1) In the Introduction, please give a clear definition of sarcopenia.
2) Since this is a retrospective study, did all patients underwent routine CT for SMI estimation? Indeed this is not a widespread test in routine clinical practice.
3) Page 3: RECICT - - > RECIST.
4) A parameter that has not been considered in the analysis is the presence of esophageal varices. Indeed bleeding from varices is one of the most relevant causes of cirrhosis.
5) In table 3, BCLC and Child Pugh should be added in the uni/multivariate analysis to confirm that SMI is really an independent predictor for survival, other than the stage of liver disease itself.
6) Please discuss pros and cons of CT evaluation od SMI in cirrhotic patients, compared to other techniques (see Barone M et al, Nutrition 2022).
Comments on the Quality of English LanguageSee above
Author Response
Response to Reviewer #2
We are pleased that in the overall comments this reviewer found our study is of interest. We also thank this reviewer’s constructive comments which were most helpful to improve our manuscript. We accordingly revised the manuscript as follows.
- In the Introduction, please give a clear definition of sarcopenia.
According to this suggestion, we added a clear definition of sarcopenia in the Introduction section (Page2, lines 32-33). Thank you for your kind suggestion.
- Since this is a retrospective study, did all patients underwent routine CT for SMI estimation? Indeed this is not a widespread test in routine clinical practice.
In our hospital, dynamic computed tomography (CT) is performed every 3 or 4 months during chemotherapy to evaluate treatment response in each case. We added this surveillance strategy in our hospital in the Materials and Methods section (Page 3, lines 5-7). Thank you again for your kind suggestion.
- Page 3: RECICT - - > RECIST.
According to this indication, we revised this word. (Page 3 lines 7).
- A parameter that has not been considered in the analysis is the presence of esophageal varices. Indeed bleeding from varices is one of the most relevant causes of cirrhosis.
In our hospital, esophagogastroduodenoscopy was conducted annually in patients with esophageal varices and endoscopic variceal ligation was performed for esophageal varices with more than F2 and/or a red color sign. Patients with no/F1/F2/F3/post EVL esophageal varices at the time of the treatment introduction were 23/5/0/0/7 for AB group, and 29/9/1/2/8 for LEN group, respectively. Only one patient (LEN group) experienced bleeding from varices. These facts were noted in the Results section (Page 4, lines 5-8) and in the revised Table 1.
- In table 3, BCLC and Child Pugh should be added in the uni/multivariate analysis to confirm that SMI is really an independent predictor for survival, other than the stage of liver disease itself.
According to this suggestion, we added BCLC and Child-Pugh score in Table3. The revised Table 3 showed that BCLC was not a significant predictor for survival in the univariate analysis. As for Child-Pugh score, it turned out to be one of the significant predictors for survival in the univariate analysis. However, neither ALBI nor Child-Pugh score was an independent predictor in the multivariate analysis (as shown in Table3_for_reviewer#2), which was impossible. This wrong result must arise from the extremely strong correlation between ALBI and Child-Pugh score and we decided not to adopt Child-Pugh score in the revised Table 3. SMI remained to be one of the independent predictors for survival, regardless of whether Child-Pugh score was adopted or not. We appreciate your valuable suggestion.
- Please discuss pros and cons of CT evaluation od SMI in cirrhotic patients, compared to other techniques (see Barone M et al, Nutrition 2022).
According to the previous study this reviewer indicated, several methods are used in clinical practice to measure body composition, including skinfold thickness assessment, bioelectrical impedance analysis, dual-energy X-ray absorptiometry, CT (used in this study), and quantitative magnetic resonance imaging. Among them, we believe that CT is the most suitable assessment technique for patients with HCC because it is already commonly used to evaluate the treatment response for HCC. We added these descriptions in the Discussion section with a new citation #25 (Page 8, lines 26-31). These revisions contribute significantly to the paper's quality. We appreciate your valuable suggestion again.

Round 2
Reviewer 2 Report
Comments and Suggestions for Authors
All answers were ok